# Combining the Anticancer and Immunomodulatory Effects of *Astragalus* and Shiitake as an Integrated Therapeutic Approach

**DOI:** 10.3390/nu13082564

**Published:** 2021-07-27

**Authors:** Biju Balakrishnan, Qi Liang, Kevin Fenix, Bunu Tamang, Ehud Hauben, Linlin Ma, Wei Zhang

**Affiliations:** 1Centre for Marine Bioproducts Development, College of Medicine & Public Health, Flinders University, Bedford Park, Adelaide, SA 5042, Australia; bbal015@aucklanduni.ac.nz (B.B.); qi.liang@flinders.edu.au (Q.L.); tama0054@flinders.edu.au (B.T.); 2The Basil Hetzel Institute for Translational Health Research, Adelaide, SA 5011, Australia; kevin.fenix@adelaide.edu.au; 3Shanxi University of Traditional Chinese Medicine, Taiyuan 030600, China; 4Discipline of Surgery, Adelaide Medical School, Faculty of Health and Medical Sciences, The University of Adelaide, Adelaide, SA 5000, Australia; 5AusHealth Corporate Pty Ltd., Adelaide, SA 5032, Australia; 6Griffith Institute for Drug Discovery, Griffith University, Brisbane, QLD 4111, Australia

**Keywords:** *Astragalus*, Shiitake, Hengshan *Astragalus* Shiitake, cancer immunology, medicinal food, bioactive

## Abstract

*Astragalus* root (Huang Qi) and Shiitake mushrooms (*Lentinus edodes*) are both considered medicinal foods and are frequently used in traditional Chinese medicine due to their anticancer and immunomodulating properties. Here, the scientific literatures describing evidence for the anticancer and immunogenic properties of Shiitake and *Astragalus* were reviewed. Based on our experimental data, the potential to develop medicinal food with combined bioactivities was assessed using Shiitake mushrooms grown over *Astragalus* beds in a proprietary manufacturing process, as a novel cancer prevention approach. Notably, our data suggest that this new manufacturing process can result in transfer and increased bioavailability of *Astragalus* polysaccharides with therapeutic potential into edible Shiitake. Further research efforts are required to validate the therapeutic potential of this new Hengshan *Astragalus* Shiitake medicinal food.

## 1. Introduction

Cancer cases are increasing worldwide [1]. In Australia, about 400 new cancer cases are diagnosed daily, with 140 associated deaths every day [2]. Cancer is a complex multifactorial group of diseases involving numerous pathogenic pathways of tumorigenesis and prognosis [3]. Studies addressing the complexity of cancer have made it clear that cancer prevention and treatment will require a combination of lifestyle changes and preventative measures, as well as diagnostic and therapeutic strategies [4,5]. Following early detection, certain cancers can be effectively treated with surgery followed by radiotherapy or chemotherapy. However, many current treatments produce systemic toxicity resulting in severely reduced quality of life [6,7,8], strongly reflected in younger women undergoing treatment for gynecological cancer, which reduces their fertility [9]. This necessitates the development of safe preventative and targeted therapeutic technologies.

Dietary and metabolic factors play important roles in the induction, prevention, and treatment of cancer [10,11,12]. In a recent study, researchers have identified a positive association of a high-fat cheese diet and an inverse association of fiber-rich food with the incidence of colon cancer [13]. Dietary components that provide health benefits and essential nutrients in a whole, fortified, enriched or enhanced form are called medicinal foods [13]. The capacity of various medicinal foods to mediate anticancer activity has been studied in a variety of cancers both in vitro and in vivo [14,15,16,17,18]. Medicinal foods have also been shown to improve the quality of life in women during the menopausal stage [19,20]. Indeed, traditional medicinal foods such as seaweed polysaccharides were shown to effectively reduce gut dysbiosis [21,22]. Therefore, there is a great medical need and commercial interest to determine the therapeutic value of medicinal foods used in Traditional Chinese Medicine (TCM). Bioactive compounds from TCM plants were shown to improve the quality of life of cancer patients when combined with chemotherapy or immunotherapy [23,24,25,26]. In certain cases, these beneficial effects have been attributed to immune-activating properties [27,28].

*Lentinus edodes* (Shiitake) and *Astragalus membranaceus* (Huang Qi) are two representative types of medicinal food that have been broadly studied for their anticancer and immunologic effects. In this paper, we reviewed the available literature (2010 to 2021) by searching the databases PubMed, Scopus, Google Scholar, Medline and Web of Science on the anticancer and immunogenic properties of Shiitake mushrooms and *Astragalus* root (Appendix A). Based on our experimental data, we also assessed the potential to develop medicinal food with combined bioactivities using Shiitake mushrooms grown over *Astragalus* beds (Hengshan *Astragalus* Shiitake, HAS) in a proprietary manufacturing process, as a novel cancer prevention approach.

## 2. Anticancer Effects of *Astragalus* and Its Main Medicative Compounds/Extracts

*Astragalus* is a medicinal herb used in TCM due to its anticancer and immunomodulating properties [29,30]. *Astragalus* is rich in bioactive substances, including polysaccharides, flavonoids and triterpene glycosides [30]. *Astragalus* polysaccharides (APS) are mainly extracted from roots and stems [31] and consist of heteropolysaccharides, composed of nine monosaccharides: glucose, arabinose, mannose, galactose, xylose, rhamnose, ribose, fucose and fructose [32,33]. Notably, the purified APS were shown to significantly inhibit the proliferation of breast cancer cells in a dose-dependent manner. This inhibitory effect could be reversed by the wnt/β-catenin agonist lithium chloride (LiCl), suggesting that the anticancer activity of APS is mediated through blocking the canonical wnt/β-catenin signaling pathway [34]. Similar effects were also observed in gastric, lung and hepatic cancer cell lines [35,36,37,38].

In addition to their direct antiproliferative effects on cultured cancer cells, APS was also shown to inhibit epithelial to mesenchymal transition (EMT), a process by which epithelial cells gain invasive and migratory characteristics of mesenchymal stem cells [39]. Mesenchymal stem cells, also known as mesenchymal stromal cells, can promote many types of cancers due to their angiogenesis and immune evasion capacity [40]. To investigate the effect of APS on mesenchymal cell-dependent lung cancer growth, researchers co-cultivated bone marrow-derived mesenchymal cells with A549 lung cancer cells, in the presence or absence of APS. Notably, treatment with APS resulted in a reduced proliferation, absence of EMT, and cell cycle arrest in these cancer cells. Furthermore, APS inhibited mesenchymal cell-induced expression of RAS/ERK and MAPK/NFkB pathway-related proteins in cancer cells, supporting the potential use of *Astragalus* to treat non-small-cell lung cancer [39].

APS is the best-studied bioactive in *Astragalus*, but not the only one. Another major group of chemicals present in *Astragalus* with potential anticancer activity consists of saponins, a large family of structurally related compounds containing a steroid or sapogenin linked to one or more oligosaccharide moieties. Guo et al. found that *Astragalus* saponins (AST) activate pro-apoptotic pathways in colorectal cancer (CRC) cell lines as indicated by a decrease in the expression of anti-apoptotic proteins Bcl-2, PARP and pro-caspase 3 and an increase in the expression of pro-apoptotic proteins Bax, Bak and Bad [41]. Adding AST in the drinking water of mice with DSS-induced CRC resulted in significant tumour growth reduction [41]. Furthermore, AST treatment attenuated the inflammatory response and the Warburg effect in tumour cells [41]. In oncology, the Warburg effect refers to a modified form of cellular metabolism found in most cancer cells, where the cells predominantly produce energy by a high rate of glycolysis followed by lactic acid fermentation, rather than by a comparatively low rate of glycolysis followed by oxidation of pyruvate in mitochondria as in most normal cells [42]. The authors hypothesized that the AST inhibits the Warburg effect by downregulating the expression of the c-Myc oncogene, which was reported to induce the glycolytic pathway, and thereby produce tumour-associated inflammation. This study is important since it not only shows the pro-apoptotic mechanism of AST but also AST’s action on the Warburg effect. These authors also suggest potential application of AST in the prevention of tumor-associated inflammation in addition to its anticancer mechanisms.

One of the members of the AST family is Astragaloside IV, which is a cycloartane-type triterpene glycoside compound. Different research teams have demonstrated the beneficial effects of Astragaloside IV against various tumour types. For example, Astragaloside IV was shown to suppress the tumour growth of lung cancer [43] and breast cancer [44] in animal models, as well as reduce cancer cell viability in hepatocellular carcinoma [45] and gastric cancer [46]. In addition to its antiproliferative effects, Astragaloside IV can also inhibit tumour cell migration and invasion, enhance chemosensitivity of cancer cells when used as an adjuvant with chemotherapy [47], reverse multidrug resistance [48] and promote immune response against cancer cells [45]. 

Flavonoids are a class of polyphenolic secondary metabolites composed of antoxanthins, flavanones, flavanonols, flavans, autocyanidins, and isoflavonoids, which exist extensively in plants. Flavonoids are not only the most important plant pigments, but also chemical messengers, physiological regulators and cell cycle inhibitors [49]. For humans, these compounds have been demonstrated to have a variety of biological and pharmacological effects, including antimicrobial, antihypertensive, antioxidant, antiaging, antifatigue and anti-inflammatory activities as well as liver protection, immune enhancement, radiation protection and lipid profile improvement functions [50,51,52,53].

Flavonoids have also been demonstrated to have multiple anticancer activities such as inhibiting cancer cell proliferation, inducing apoptosis, suppressing the secretion of matrix metalloproteinases, restraining angiogenesis and preventing tumour invasive behaviour [54]. Flavonoids from *Astragalus* complanatus (FAC) could inhibit the proliferation of breast cancer cells in a dose-dependent manner and suppress the cancer cell invasion in vitro, while in vivo administration of FAC increased the survivability of xenografted nude mice models of breast cancer [55]. These anticancer effects may be attributed to the promotion of apoptosis and regulation of metastasis-related gene expression by FAC [55]. Similarly, flavonoids of *Astragalus* also showed inhibitory effects on human hepatocellular erythroleukemia cells [56] and carcinoma cells [57] by arresting the cell cycle at G0/G1 phase.

## 3. Anticancer Effects of Shiitake and Its Main Medicative Compounds/Extracts

Shiitake (*Lentinus edodes*) has been shown to share some of the anticancer, and immunomodulatory effects of *Astragalus*, although not as potent [58]. The medicinal properties and oral bioavailability of Shiitake, however, have made it a popular dietary product consumed broadly worldwide and used in TCM for thousands of years [24,28,58]. 

The major anticancer and pro-apoptotic activities of Shiitake come from its β-glucans, a group of β-d-glucose polysaccharides [59,60]. Sari et al. tested the effects of Shiitake-derived β-glucans on triple-negative breast cancer cells and non-small-cell lung cancer cells and found that the molecular weight of β-glucans is a determinant factor for their anticancer effects [60]. Interestingly, low molecular weight β-glucans (5 KDa) did not show any significant anti-proliferation effects on cancer cells, while cell viability was reduced by >90% by treatment with high molecular weight (>300 KDa) β-glucans, likely due to induction of apoptosis [60].

One of the best-characterized polysaccharides in Shiitake is a water-soluble compound known as lentinan. Its chemical structure consists of a β-1, 3 glucopyranoside main chain with β-1, 6 glucopyranoside branched chain [61]. This β-glucan structure of lentinan was shown to mediate anticancer effect in vitro and in vivo [62]. There is evidence to suggest that the structural conformation of lentinan is critical for its anticancer activity; Indeed, the triple-helix conformation of lentinan had a higher anticancer effect compared to its single flexible chain structure [63]. Three different β-glucans with β-(1→6), β-(1→3), (1→6) and α-(1→3) linkages purified from polysaccharide-rich extracts were tested for their hypocholesterolemic, anticancer, anti-inflammatory and antioxidant activities in vitro [64]. These β-glucans reduced the secretion of IL-1β and IL-6 from LPS-activated THP-1 cells and selectively induced cell death of breast cancer cells while not affecting normal breast cells [64]. The results of this study also demonstrate that d-glucans isolated from Shiitake have medicinal properties including anticancer activity. A recent review summarized evidence for potential use of lentinan in the treatment of many cancer types including lung cancer, gastric cancer, colorectal cancer, ovarian cancer, cervical cancer, and Non-Hodgkin lymphoma [64]. Notably, clinical studies have shown that lentinan improves the quality of life of cancer patients [64].

Another major bioactive protein present in Shiitake is Latcripin. The antiproliferative activities of Latcripin-1 were tested on six cancer cell lines (SGC-7901, BGC-823, SKOV-3, HepG-2, MDA-MB-231, MCF-7) and two normal cell lines (GES-1 and HaCaT). Gastric cancer cells (SGC-7901 and BGC-823) were found to be the most sensitive to Latcripin-1 [65], mainly due to induction of cell cycle arrest at S phase and the induction of autophagy, but also attributed to the reduced MMP-2/9 levels responsible for gastric cancer migration and invasion. Furthermore, Latcripin-1 treatment caused a dose-dependent decrease in the expression of the antiapoptotic protein Bcl-2 and an increase in pro-apoptotic proteins Bax and Caspase-3. Latcripin-1 treatment also suppressed Akt/mTOR phosphorylation and PI3K expression, which are critically involved in cancer progression. This suggests that Latcripins may have the capacity to reduce cancer cell survival, invasion and migration [65]. Similar to Latcripin-1, Shiitake-derived Latcripin-4 was also shown to inhibit the cell viability the HepG2 cells [66], while Latcripin 11 had anticancer activity on eight cancer cell lines without causing any toxicity to HUVEC (Human Umbilical Vein Endothelial cells) [67]. The results of these studies support the use of Latcripins in cancer prevention and therapy, therapeutic and hence the high potential of Shiitake as medicinal food.

A popular avenue currently being explored is whether there is synergy in combining different medicinal foods together. Investigation of the anticancer activity of the mycelial extracts of Shiitake in combination with specific nutrients or other medicinal mushrooms has revealed positive findings. For example, a combination of the standardized cultured extract of Shiitake mycelia (also known as Active Hexose Correlated Compound, AHCC) and *Wasabia japonica* (Wasabi) (combinedly called Bioactive Immunomodulatory compound, BAIC) was tested for its effects on breast and pancreatic cancer cells [68]. The results demonstrated that BAIC at very low concentrations reduced the viability of cancer cells more than either AHCC or Wasabi used alone. The anticancer activity of BAIC was mediated through increasing the G0/G1 population and augmenting the percentage of apoptotic cells by upregulating the expressions of pro-apoptotic proteins [68]. A study exploring the synergistic effects of combining the mushroom *Agaricus bisporus* and Shiitake (*Lentinus edodes*) on prostate cancer cells demonstrated significant reduction of IL-8, NFkB and nuclear to cytoplasmic ratio, suggesting an enhanced anticancer efficacy of this combined formulation [69]. Tumour suppressive effects were also achieved in vivo by using a novel mushroom blend (Micotherapy U-care), which was shown to be effective in reducing the tumour burden in a triple-negative breast cancer mouse model [70]. These studies support the promising potential of shiitake in combined formulations for cancer therapy. The major pathways through which bioactives in *Astragalus* and Shiitake cause antiproliferative /pro-apoptotic effects are summarized as in the Figure 1. 

## 4. The Potential Use of *Astragalus* and Shiitake as Adjuvants in Chemotherapy

In addition to its therapeutic effects, chemotherapy also induces severe adverse effects such as hair loss, fatigue, infection, nausea and vomiting, which significantly compromised the quality of life of cancer patients. Besides their direct anticancer effects, a highly valuable function of medicinal food is that they can enhance the chemosensitivity of the cancer cells and therefore reduce effectively the doses of chemodrugs in a therapeutic regimen. The anticancer properties of plant and mushroom extracts in combination with standard chemotherapeutic agents are summarized in Appendix A. For example, co-administration of Astragaloside IV with cisplatin in a mouse model enhanced chemotherapeutic activity of cisplatin against hepatocellular carcinoma, as evidenced by delayed tumour formation and more apoptotic areas in the tumors compared to cisplatin treatment alone [71]. Further analysis revealed that the MRP2 (Multidrug resistance-associated protein 2) expression was significantly reduced in hepatocellular cancer cells that received a combinational treatment. Moreover, the authors reported that combination therapy induced a nephroprotective effect by the ability of Astragaloside IV to reduce the release of inflammatory mediators such as creatinine and blood urea nitrogen [71]. 

APS could also act synergistically or additively with standard chemotherapeutic agents, therefore reducing the dosing of standard chemotherapy and minimizing toxic side effects. In a recent study, APS has been shown to promote apoptotic effects of Adriamycin on gastric carcinoma cells when used in combination with Adriamycin [72]. The results of this study indicate that the mechanism for the anticancer activity was mediated through the AMPK pathway and APS can synergistically act with Adriamycin to enhance its anticancer activity [72]. In another study, Apatinib (a tyrosine kinase inhibitor used in many cancer targeted therapies), when used in combination with APS, showed enhanced anticancer effect on pancreatic cancer cells by reducing cell invasion and migration significantly compared to Apatinib monotherapy [73]. The combination therapy significantly downregulated the phosphorylated protein kinase B (AKT), extracellular signal-regulated kinase (ERK) and MMP-9 expression and mediated anticancer activity in a synergistic/additive manner [73].

The adjuvant activity of lentinan on cancer treatment has also been widely reported [64,74]. Importantly, lentinan injection was approved as an adjuvant chemotherapy for gastric cancer patients in Japan in 1985 [74]. It has also been used as an adjuvant with paclitaxel and cisplatin for gastric cancer [75,76]. Similarly, combination therapy of lentinan with other drugs such as OK-432 (picibanil) and docetaxel has been proven effective in the treatment of gastric cancer without serious adverse reactions [77,78]. For hepatoma, lentinan in combination with cisplatin showed promising synergistic antitumor action and diminished adverse side effects [79]. In this study, the authors explored the molecular mechanisms of the observed synergistic effects and found that they resulted from Caspase-3, -6, -7 and -8 activation, increased Bax/Bcl-2 ratio and decreased expression of Stat-3 and NF-κB [79]. Furthermore, in a clinical trial, patients receiving S-1-based fluoropyrimidine and lentinan co-treatment had a significantly longer median overall survival compared to those who received chemotherapy alone [80]. In the combined treatment group, the ratio of granulocytes to lymphocytes in patients was also higher, suggesting that the nonspecific activation of macrophages by Lentinan might have skewed the T helper (Th) phenotype towards Th1 and resulted in Th1 mediated oncolytic activity. These authors suggested that this can be a main contributing factor underlying the prolonged survival in the lentinan adjuvant group [80]. The major adjuvant activity of combination therapy with *Astragalus* extracts is shown in Table 1. 

## 5. Immunomodulatory Functions of *Astragalus* and Shiitake in Tumour Immune Microenvironment

Various cancer types are characterized by widespread immunosuppression including compromised immune surveillance mainly by downregulation of tumour cell expression of major histocompatibility complex (MHC) class I molecules, which are a group of cell surface proteins presenting tumour-specific antigens to adaptive immune cell that can eliminate cancer cells [81,82]. Other mechanisms that contribute to immune evasion include the failure of dendritic cells, the professional antigen-presenting cells that infiltrate tumors, to present tumour antigens, expression of immune checkpoint inhibitors by tumour cells, and the inability of effector T cells to mediate cytotoxic responses within the tumour microenvironment [83]. Boosting effective antitumor immunity has therefore become a major therapeutic strategy against cancer.

Both in vitro and in vivo studies have shown that *Astragalus* and Shiitake can enhance immune responses in cancer therapy (Appendix A). Following co-administration with 5-FU, APS significantly enhanced the proliferation of splenocytes and increased phagocytosis by peritoneal macrophages in mice, resulting in immune activation and inhibition of tumour growth [84,85]. Tumour infiltrating macrophages can be subdivided into tumour-suppressing M1 macrophages and tumour-promoting M2 macrophages [86]. Notably, in vivo treatment with APS was shown to induce macrophage’s polarization to M1 phenotype via the Notch signaling pathway, thereby reducing the cancer progression [87]. Similarly, a study investigating the effects of APS (PG2) on anticancer immunogenicity showed that PG2 significantly increased M1/M2 macrophage ratio, which was positively correlated with the downregulation of tumour-promoting cytokines such as IL-6 and IL-10 [29]. PG2 also promoted DC maturation with concomitant T cell-mediated anticancer immune response [29]. In an in vitro explant culture activity, APS treatment was shown to have significant inhibition on Tregs (Regulatory T Lymphocytes) functions. The inhibitory effect was dose/time-dependent, with most significant changes observed at 48 h and at a APS concentration of 100 µg/mL. At this concentration, the levels of IFN-_Ƴ_, IL-10 and IL-4 in supernatant and malignant tissues were reduced in comparison to the untreated control group. Further, APS at 100 µg/mL reduced the proliferation of CD4^+^CD25^+^ cells. APS treatment has reduced the expression of FOXP3 mRNA in supernatants of CD4^+^CD25^+^ cells in a dose- and time-dependent manner. FOXP3 functions as a master regulator of the signaling pathway in the development and function of regulatory T cells, which can prevent the immune system from destroying cancer cells when they are in excess. The migratory function of CD4^+^CD25^+^ Tregs was also inhibited by APS, thus inhibiting Th2 cytokine expression and stimulating cytotoxic response mediated by Th1 cells. In addition, these authors observed a reduction in secretion of the chemokine CXCL12. Based on this observation, they proposed that the inhibitory effects of APS on migration of CD4^+^CD25^+^ Treg cells is correlated with the CXCR4/CXCL12 signaling pathway [88]. Moreover, APS has also been shown to boost the phagocytic capacity of peritoneal macrophages and the proliferation of splenic lymphocytes; increase the serum antibody titters and the production of cytokines such as IL-4 and IL-10, CD40, CD86 in dendritic cells; and promote the maturation of dendritic cells and the transformation rate of peripheral blood neutrophils and lymphocytes [89].

In Asia, lentinan has been used as an immune booster for thousands of years [90]. Lentinan treatment on phorbol 12-myristate 13-acetate (PMA) differentiated THP-1 macrophages induced IL-10 expression at higher levels than LPS, mediated by the expression of dectin-1, a receptor that recognizes β-glucans on THP-1 cells [91]. This suggests that β-glucans may have an immunomodulatory function. The findings from in vitro studies are largely supported by in vivo research using disease animal models. For example, the effect of lentinan on inflammasome activation was studied in a mouse model, where lentinan was shown to selectively inhibit interferon-inducible protein AIM2 (absent in melanoma 2)-mediated inflammasome activation and to upregulate proinflammatory cytokines through toll-like receptor 4 signaling [92]. Furthermore, mice fed with Maitake (*Grifola frondosa*), Shiitake or a combination of both had enhanced phagocytic ability mediated by peritoneal monocytes and neutrophils, enhanced NK cell activity and boosted serum levels of proinflammatory cytokines IL-6, IL-12 and IFN-γ compared to controls [93]. Interestingly, combined oral administration of polysaccharide extracts from Shiitake and *Astragalus* was shown to improve cell- and humoral-mediated immune response with a potential to enhance vaccine efficacy in chickens [94,95].

Given the promising findings from preclinical studies, the immunomodulatory potential of Shiitake was further investigated in many clinical trials. In healthy individuals, consumption of either 5 g or 10 g shiitake per day for 14 days resulted in improved immune functions as measured by increased activation in ɣδ-T cells, NK cells and Natural Killer T lymphocytes (NK-T cells) [96]. The amount of secretory immunoglobulin A (sIgA) in saliva was higher and C-reactive protein (CRP) was lower after consuming shiitake. Ex vivo stimulation of NK-T cells with mitogens, cytokines such as IL-4, IL-10, IL-1α and TNF-α had significantly higher levels, MIP-1α/CCL3 secretions were significantly lower, while the levels of IL-1β, IL-17, IFN-ɣ and MIP-1β were unaltered [96]. However, when less shiitake (700 mg twice a day) was consumed in another clinical trial, no effects on immune cell numbers nor inflammatory mediators were detected in healthy volunteers undergoing exercise [97]. A more efficient way to modulate immune functions might be using purified shiitake glucans instead of unprocessed shiitake. Indeed, in another double-blinded, crossover, placebo-controlled trial of Lentinex^®^, a trademarked commercially sold product consisting of shiitake β-glucans, elderly subjects given 2.5 mg/day of Lentinex^®^ had increased numbers of B cells and NK cells, demonstrating immunostimulatory activity of Lentinex^®^ in healthy elderly subjects [98].

In lung cancer patients, lentinan combined with chemotherapy was shown to improve quality of life and immunological endpoints when compared to patients given chemotherapy alone [90]. A pilot study assessing the safety of Shiitake mycelia extract as adjuvant therapy in cancer patients found in patients receiving two treatment courses a significant improvement of quality of life, enhanced lymphokine-activated killer (LAK) cell and NK cell activity, and reduced serum immunosuppressive acidic protein levels [99]. However, this study was not a randomized control study [99]. Another randomized, double-blinded, placebo-controlled study suggested that patients who received Shiitake mycelia (*Lentinus edodes* mycelium-LEM) along with anthracycline-based chemotherapeutic agents had their quality of life and immune parameters improved. The LEM-treated group had lower frequency of Tregs, indicating the presence of more effector CD4+ T helper cells [100]. Combinational chemotherapy using lentinan has also been suggested to be effective in improving the overall survivability of cancer patients with extensive liver metastasis [101]. In this study, two patients with multiple liver metastases and high human epidermal growth factor receptor 2 (HER 2) expression were treated with trastuzumab (humanized monoclonal antibody specific for HER2/Neu Receptor) and showed improved responses in overall survival when compared with those who only received chemo-immunotherapy. Mechanistically, the authors postulated that lentinan effects on leukocytes could enhance the antitumor effect of trastuzumab by augmenting antibody-dependent cellular cytotoxicity (ADCC) [101]. In another clinical study, the effect of combined treatment of LEM on dendritic cell-based cancer vaccine therapy or CD3-activated T-lymphocyte (CAT) therapy was assessed in 10 cancer patients [102]. Significant effects on the quality of life and immune parameters were observed in the patients, as demonstrated by an improvement in quality-of-life scores and increased IFN-ɣ production by patient PBMCs, which was shown to correlate with changes in Tregs (such as increased expression of FOXP3 and CD4+ and increased secretion of TGFβ) [102]. The immunomodulatory functions of *Astragalus* and Shiitake on tumour microenvironment are summarized in the Figure 2. 

## 6. Hengshan *Astragalus* Shiitake (HAS)-A Novel Shiitake-Based Medicinal Food

The above literature supports the notion that specific compounds in both *Astragalus* and Shiitake have different anticancer and immunomodulatory properties. In addition, *Astragalus* has lower appeal as a medicinal food compared to Shiitake due to attributes related to taste and other attributes. Therefore, it can be beneficial to combine Shiitake and *Astragalus* into one medicinal food product with culinary appeal and promising health benefits (Figure 3). Indeed, many commercial formulations of this combination are currently available in the market (Reishi Shiitake *Astragalus* Compound, Ki Immune Defence & Energy formula, Mushroom Immunity, etc.). However, all the available products are formulated as a drug supplement, without the natural benefits of Shiitake as a popular and delicious food.

To fill this gap, a new product, Hengshan *Astragalus* Shiitake (HAS), was developed through a proprietary manufacturing process to grow Shiitake on cultivation beds containing *Astragalus* from Hengshan, China. Here, we performed a metabolomic analysis to test whether bioactive components have been transferred from the *Astragalus* cultivation bed into Shiitake, using this novel manufacturing process.

### 6.1. HAS Cultivation

HAS cultivation was carried out by Shanxi Yulongxiang Agricultural Development Co Ltd. in China. The protocol includes cultivating Shiitake, on the beds of residues containing the medicinal part of Hengshan *Astragalus*. Briefly, the HAS cultivation process involves the preparation of HAS growth medium, inoculation and germination, fruiting picking and air-drying. After removing the medicinal part of *Astragalus* roots, the residues were used at 5% and 20% of the total bed (HAS growth medium) for cultivation of Shiitake. The mushrooms obtained from this cultivation processes are shown in Figure 4.

### 6.2. Metabolomic Analysis of HAS

To characterize the differential molecular composition of Hengshan *Astragalus* (HA), Shiitake (S), HAS-5% and HAS-20%, a global metabolomics study was carried out using GC/LC/MS/MS techniques. Standard ethanolic extraction was used to analyze the secondary metabolites. Briefly after deproteination, the supernatants were transferred to fresh tubes. A derivatization procedure using Bis(trimethylsilyl)trifluoroacetamide-1% Trimethylchlorosilane (BSTFA) regent (1% TMCS, v/v) was carried out to improve the detection limit. QC samples were similarly prepared. All samples were then analyzed by a gas chromatograph coupled with a time-of-flight mass spectrometer (GC-TOF-MS) (Agilent 7890 GC-TOF- MS) or UHPLC-QTOF-MS (Ultra-High-Performance Liquid Chromatography Time-Of-Flight Mass Spectrometry). The data analysis, including peak extraction, baseline adjustment, deconvolution, alignment and integration, was carried out with Chroma TOF (V 4.3x, LECO) software and LECO-Fiehn Rtx5 database was used for metabolite identification by matching the mass spectrum. Retention index or MS raw data (.wiff) files were converted to the mzXML format by ProteoWizard and processed by R package XCMS (version 3.2). The process includes peak deconvolution, alignment and integration. Minfrac and cutoff were set as 0.5 and 0.3, respectively. In-house MS2 database was applied for metabolites identification. Finally, if the peaks were detected in less than half of the total QC samples, those peaks were removed.

In total, 3307 peaks were obtained in the negative mode, of which 556 were detected with MS/MS, 1823 were detected with MS, while 928 were unmatched with any compound in the library. Similarly, 3130 peaks were obtained in the positive mode, of which 851 were detected with MS/MS, 1407 were detected with MS and 872 did not match any of the compounds in the library. The total ionic current for the peaks obtained is shown in Figure 5A and differential expression of unique metabolites is shown in the Venn diagram (Figure 5B) and in Table 2. From these MS traces (Figure 5A), it is evident that Shiitake, HAS-5% and HAS-20% share many major secondary metabolites, although the concentrations of these metabolites are higher in HAS-5%. On further evaluations, there appeared to be some metabolites from *Astragalus* that were present in HAS-5% and/or HAS-20%, but absent from Shiitake. As shown in Figure 5B and Table 2, 11 metabolites, namely aspargine 4, 3-cyanoalanine, 2-deoxytetronic acid, erythrose 2, l-threose 2, 2,4-diaminobutyric acid 3, 3-methylamino-1,2-propanediol 2, cholestan-3beta-ol, and (2r,3s)-2-hydroxy-3-isopropylbutanedioic acid, were successfully transferred from HA to HAS-5% and HAS-20%. Interestingly, cholestane-3,5,6-triol and 1,4-cyclohexanedione 1 were only detected in HAS-5%, while cycloleucine 1 and resveratrol 1 are unique metabolites detected only in HAS-20%. Some metabolites transferred from HA to either HAS-5% or HAS-20% are unidentified due to unavailable information in the annotated metabolite libraries. The results of our metabolomics analysis provide proof of concept for the hypothesis that the cultivation of Shiitake on the optimized *Astragalus* bed leads to transfer of secondary metabolites from *Astragalus* to Shiitake.

The therapeutic potential of the transferred metabolites and the anticancer function or immunogenic activity of HAS products remains to be elucidated. However, several of these transferred metabolites (Table 2) have been demonstrated to have medicinal functions. For example, threose nucleic acids were found to inhibit anti-BcL-2 (anti-apoptotic protein) by specifically targeting both mRNA and protein expression of BcL-2 in cancer cells [103]. Another compound, Asparagine, which has been transported from HA to HAS, was shown to have antiproliferative activity and to regulate mTOR signaling in cancer cells [104]. Similarly other transferred compounds such as Erythrose [105,106], 2,4 diaminobuturic acid [107], Cholestane-3β, 5α, 6β-triol [108] were found to have antiproliferative and antineoplastic properties.

## 7. Conclusions

Shiitake is a world-famous culinary–medicinal mushroom, rich in protein, fiber, B vitamins, vitamin D, antioxidants, potassium, and selenium; while *Astragalus* is a major medicinal herb commonly used in many herbal formulations in the practice of traditional Chinese medicine due to its well-demonstrated immunomodulating, anti-oxidant, anti-inflammatory, and anticancer effects. We conducted a literature survey to explore the anticancer and immunomodulatory properties of *Astragalus* and Shiitake. Preclinical and clinical trials have demonstrated anticancer and immunomodulatory properties of *Astragalus* and Shiitake, with *Astragalus* having stronger anticancer than immunomodulating effects, while Shiitake has more immunomodulatory properties than anticancer properties. Combination of the powerful pharmacological effects of *Astragalus* with the rich nutrition and delicious taste of Shiitake mushrooms will transform an effective medicine into a popular food product, so that more people will benefit through their daily diet. Here, a pilot experiment was carried out to study whether bioactive components shown to have anticancer effects could be absorbed and transferred from *Astragalus* to Shiitake when the Shiitake is cultivated on beds containing HAS growth medium. Indeed, our metabolomics analysis demonstrates the transfer of secondary metabolites from *Astragalus* to Shiitake, when cultivated in this manner. Future work testing the oral bioavailability and in vivo therapeutic efficacy of HAS in immunomodulation and inhibition of the proliferation of cancer cells and of tumors growth will provide a proof of concept for the efficacy and safety of such a transfer in cancer prevention and therapy. Further functional studies of HAS in immunomodulation and anticancer effects are required to establish the benefits of this cultivation process. Nevertheless, a successful outcome of this attempt to develop a new cultivation process to produce novel medicinal food can lead to a strategy to maximize the medicinal benefits of traditional medicine by improving the bioavailability and attractiveness of bioactive healthy metabolites and packaging them in delicious food, which can be consumed daily to boost health.

## Figures and Tables

**Figure 1 nutrients-13-02564-f001:**
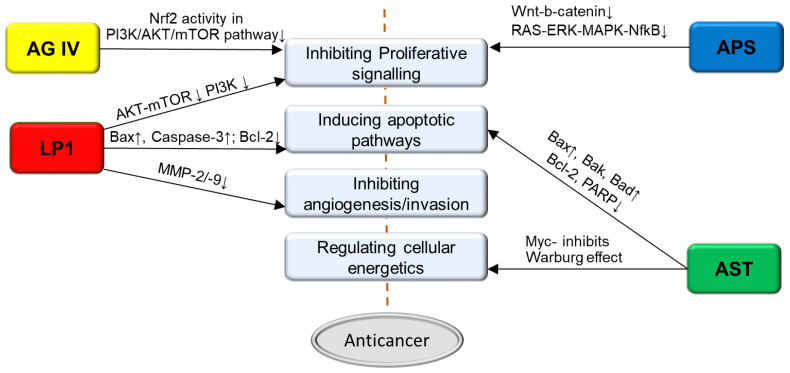
Pathways through which *Astragalus* and shiitake bioactives compounds mediate antiproliferative and pro-apoptotic effects in cancer cells. (APS–*Astragalus* Polysaccharides, AST–*Astragalus* saponins, LP1-Lacriptin domain, AGIV- Astragaloside IV).

**Figure 2 nutrients-13-02564-f002:**
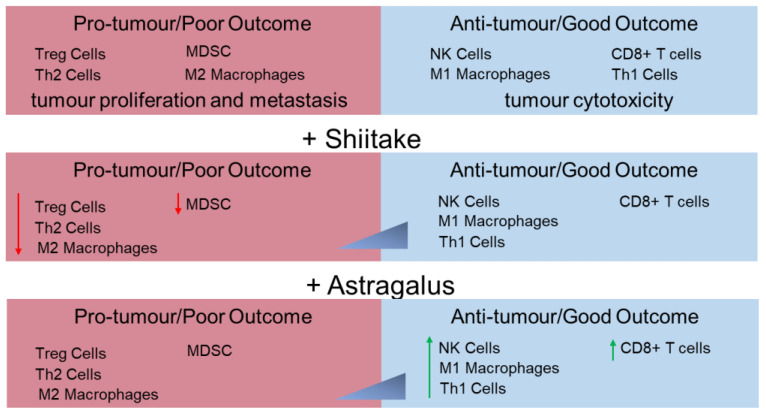
Immunomodulatory functions of *Astragalus* and Shiitake in tumour immune microenvironment. (MDSC–Myeloid-derived suppressor cells).

**Figure 3 nutrients-13-02564-f003:**
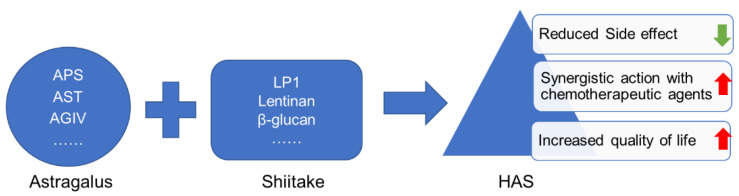
Potential benefits of HAS.

**Figure 4 nutrients-13-02564-f004:**
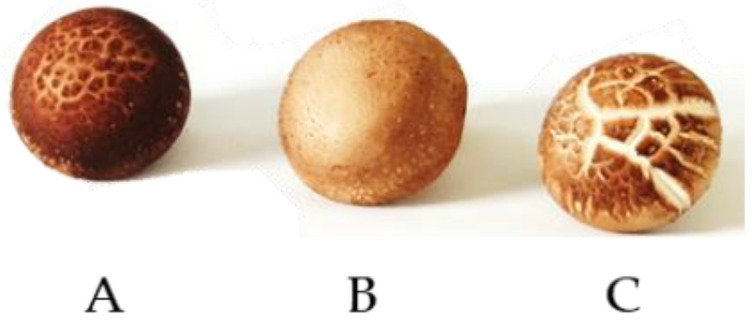
Photograph showing the (**A**) Shiitake, (**B**) HAS-5% and (**C**) HAS-20%.

**Figure 5 nutrients-13-02564-f005:**
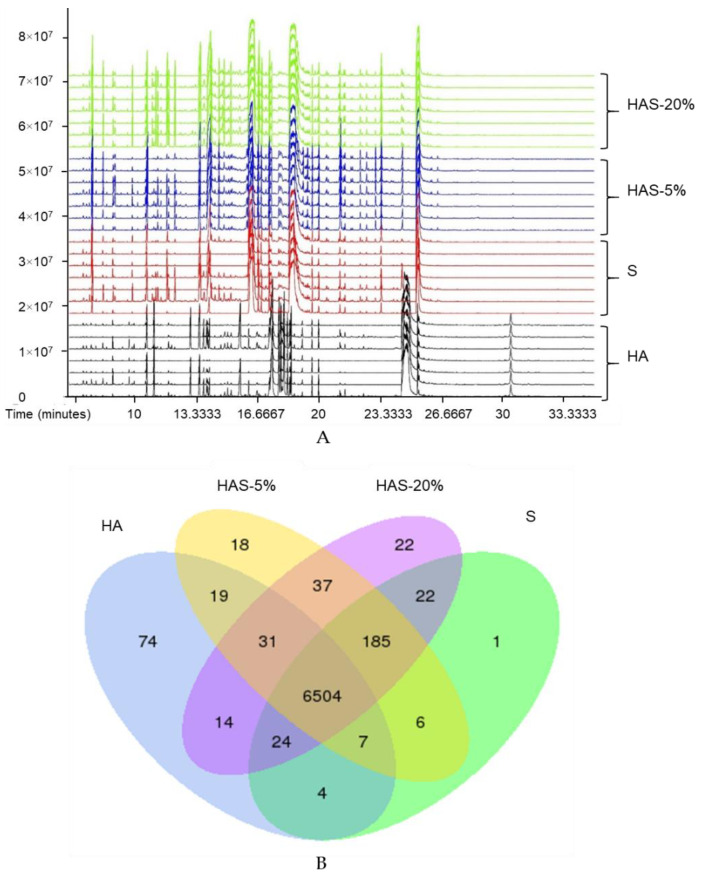
Global metabolome analysis of the differential metabolites in Hengshan *Astragalus* (HA), Shiitake (S), HAS-5% and HAS-20%. (**A**) The methanolic extracts of HA, S, HAS-5% and HAS-20% were analyzed using GC-TOF-MS with seven replicates for each sample. The peaks and signal strength of all the 28 analyses are shown as a TIC (Total Ionic Current) map. (**B**) Venn diagram showing the results from global metabolomic analysis. HA—Hengshan *Astragalus* (blue), S—Shiitake (green), HAS-5%—Shiitake grown over growth bed containing 5% *Astragalus* (yellow), HAS-20%—Shiitake grown over the growth bed containing 20% *Astragalus* (lavender).

**Table 1 nutrients-13-02564-t001:** Adjuvant activity of Shiitake/*Astragalus* extracts in combination with standard chemotherapeutic agents against cancer.

Combination Therapy	Result	Reference
Astragaloside IV + Cisplatin	Reduced cell viability and increased apoptosis in HepG2 cells	[71]
Astragaloside IV + Cisplatin	Reduced MRP-2 expression in cancer cells–lower the drug dose Reduced inflammation, thus reduces the toxicity of chemotherapeutic agent	[71]
APS + Adriamycin	Synergistic to block AMPK pathway Reduce Adriamycin dose	[72]
APS + Apatinib	Downregulating AKT-ERK pathway and MMP-9 expression–reduce invasion and migration	[73]
Lentinan + Picibanil and docetaxel	Potentiate chemotherapeutic actions against gastric cancer	[78]
Lentinan + Cisplatin	Activates pro-apoptotic pathway (Caspases 3, 6, 7 and 8)–potentiates Cisplatin’s Chemotherapeutic activity	[75]
Lentinan + Fluoropyrimidine	Increases overall survivability–potentially through reducing toxic side effects	[80]

**Table 2 nutrients-13-02564-t002:** Differential expression of metabolites in HA, S, HAS-5% and HAS-20%.

Transferred from HA to HAS-5% *	Transferred from HA to HAS-20% #	Only in HAS-5%	Only in HAS-20%
Asparagine 4	Asparagine 4	Glutamine 4	Glutamine 4
3-cyanoalanine	3-cyanoalanine	Sarcosine	Sarcosine
2-deoxytetronic acid	2-deoxytetronic acid	2-methylfumarate	2-methylfumarate
Erythrose 2	Erythrose 2	Shikimic acid	Shikimic acid
l-threose 2	l-threose 2	Maleamate 1	Maleamate 1
2,4-diaminobutyric acid 3	2,4-diaminobutyric acid 3	Tartaric acid	Tartaric acid
3-methylamino-1,2-propanediol 2	3-methylamino-1,2-propanediol 2	Capric acid	Capric acid
cholestan-3beta-ol	cholestan-3beta-ol	9-fluorenone 2	2-deoxy-d-galactose 2
(2r,3s)-2-hydroxy-3-isopropylbutanedioic acid	(2r,3s)-2-hydroxy-3-isopropylbutanedioic acid	Lyxonic acid, 1,4-lactone	Lyxonic acid, 1,4-lactone
Cholestane-3,5,6-triol	Cycloleucine 1	Acetol 1	d-glucoheptose 1
1,4-cyclohexanedione 1	resveratrol 1	Thymidine 2	Allantoic acid 3
		Glutaconic acid	Ribonic acid, gamma-lactone
		Octadecanol	Octadecanol
			Dehydroabietic acid
			Creatine degr
			n-acetyl-beta-alanine 1
			n-methyl-l-glutamic acid 2
			Salicin

* Presence in HA and HAS-5% but absent in S (potential transfer from HA to HAS-5%); # Presence in HA and HAS-20% but absent in S (potential transfer from HA to HAS-20%). Red text–common to HA, HAS-5% and HAS-20%; Blue text–common to HAS-5% and HAS-20%, but absent from both HA and S.

## Data Availability

Data available in a publicly accessible repository. The data presented in this study are openly available in FigShare at Doi.10.6084/m9.figshare.15051921.

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
