# Peer review of "Combining the Anticancer and Immunomodulatory Effects of Astragalus and Shiitake as an Integrated Therapeutic Approach"

_nutrients, 2021, doi:10.3390/nu13082564_

Round 1

Reviewer 1 Report

nutrients-1266748

In the article entitled “Combining the anticancer and immunomodulatory effects of Astragalus and Shiitake as an integrated therapeutic approach” the authors performed a review about the role of  Astragalus root and Shiitake mushroom and the promising effect of a new product called “Hengshan Astragalus Shiitake (HAS)”

The main topic of this manuscript is interesting and falls within the scope of the journal.

Some of the limitations of the study have been identified and discussed by the authors in the below section:

  • I believe English revision is needed and that probably the manuscript may benefit from some revisions
  • Introduction: Please Authors add reference to the following sentence “However, many current treatments produce high systemic toxicity resulting in severely reduced quality of life, necessitating the development of safe preventative and targeted therapeutic technologies.”
  • Introduction: Authors should underline the role of quality of life in patients affected by cancer, particularly regarding the neglected aspect of fertility preservation, briefly referring to PMID: 32419847.
  • Introduction: Please Authors better describe the literature search: what database were used?
  • Introduction: the authors discussed the role of nutraceutical foods as an interesting option in dietary pattern for patients affected by cancer. However, nutraceutical foods represent a widely practiced option in several aspects of medicine particularly in women with menopausal compliants. Authors should improve this point, at least briefly, referring to: PMID: 32693763; PMID: 31466381.
  • Conclusions: Please Authors describe the limitations of their findings regarding the role of HAS.

Reviewer 2 Report

Balakrishnan et al., in their manuscript “Combining the anticancer and immunomodulatory effects of Astragalus and Shiitake as an integrated therapeutic approach” reviewed the anticancer and immunomodulatory effects of Astragalus and Shiitake. Manuscript need to be improved based on following comments.

Major:

  1. Section 2. Anticancer effects of Astragalus was not present well. Title of this section mentioned as Anticancer effects of Astragalus, but the content has only Astragalus poly saccharides (APS) and Astragalus saponins (AST) & astragaloside IV.
  2. Provide information about other major compounds present in the Astragalus and their anticancer mechanisms.
  3. Section 2. was present in a very superficial way and not enough elements to understand the molecular mechanism. Authors just mentioned the compounds induces apoptosis in different cancer cells.
  4. Figure 1 was not informative. Redraw the figure 1.
  5. Section 5. Immunomodulatory Functions of Astragalus and Shiitake also present in a very superficial way. Provide more details like dosage & time and mechanistic pathways involved.
  6. Provide a figure illustrating the Immunomodulatory mechanisms of Astragalus and Shiitake.

Minor:

  1. Avoid formatting errors. Eg. 2. Anticancer effects of Astragalus

Round 2

Reviewer 1 Report

The authors improved the quality of manuscript as required.

I have no further concerns, and I suggest to accept this article for publication in the current form

Reviewer 2 Report

Revised manuscript might be accepted for publication.